# Predictive Minisci late stage functionalization with transfer learning

Emma King-Smith [1], Felix A. Faber [1], Usa Reilly[2], Anton V. Sinitskiy[3], Qingyi Yang[4], Bo Liu[5], Dennis Hyek[5] & Alpha A. Lee [1] ✉

Structural diversification of lead molecules is a key component of drug discovery to explore chemical space. Late-stage functionalizations (LSFs) are versatile methodologies capable of installing functional handles on richly decorated intermediates to deliver numerous diverse products in a single reaction. Predicting the regioselectivity of LSF is still an open challenge in the field. Numerous efforts from chemoinformatics and machine learning (ML) groups have made strides in this area. However, it is arduous to isolate and characterize the multitude of LSF products generated, limiting available data and hindering pure ML approaches. We report the development of an approach that combines a message passing neural network and [13]C NMR-based transfer learning to predict the atom-wise probabilities of functionalization for Minisci and P450-based functionalizations. We validated our model both retrospectively and with a series of prospective experiments, showing that it accurately predicts the outcomes of Minisci-type and P450 transformations and outperforms the well-established Fukui-based reactivity indices and other machine learning reactivity-based algorithms.

Late-stage functionalization (LSF) is a powerful technique in medicinal chemistry. The magic methyl effect describes the ability of a single methyl group, even one distal to the binding motif, to dramatically improve (or reduce) potency, solubility, and metabolic stability[1]. However, methyl groups are not the only motif that can radically change pharmacological properties. Fluoro[2], chloro[3], trifluoromethyl[4], and hydroxyl groups[5] are known beneficial motifs and/or temporary functional handles towards other beneficial motifs. Over the past several decades, numerous methods have been developed to diversify lead compounds and selectively install these biologically privileged groups directly[6–9]. One methodology commonly utilized in LSF is the Minisci-type functionalization, whereby a radical species adds to an electron-deficient (hetero)arene (Fig. 1A)[10–12]. However, the promiscuity of this single-electron method in conjunction with the inherent structural complexity of LSF molecules makes regioselectivity prediction challenging. Regiochemical predictions for Minisci-type reactions were first summarized by O'Hara et al. who developed a set of

guidelines to determine sites of reactivity based on the nucleophilicity of the alkyl radical species, pH of the reaction, solvent effects, and electronics of the heteroarene[13]. These observations were later formalized when they were noted to correlate well with the indices from Fukui functions, i.e., functions that describe the change in electron density upon the addition or removal of an electron. In the literature, Fukui-based reactivity indices predict the most reactive sites of Minisci functionalization with an average accuracy of 93% (average $F$-score of 0.77), albeit usually on smaller, minimally functionalized molecules[14,15].

There are two main approaches in the literature for regiochemical predictions: quantum chemical and data-driven. Quantum chemistry-based approaches predict reactivity and regioselectivity by computing energy barriers using techniques such as density functional theory (DFT) or machine-learning (ML) approximations of DFT-energies[16–18]. Data-driven approaches to work directly with experimental data, fitting statistical models to correlate known chemical features to real-world observed outcomes in regioselectivity[19–25]. Whilst computational data is

[1]Cavendish Laboratory, University of Cambridge, Cambridge, UK. [2]Development & Medical, Pfizer Worldwide Research, Groton, CT, USA. [3]Machine Learning Computational Sciences, Pfizer Worldwide Research, Cambridge, MA, USA. [4]Development & Medical, Pfizer Worldwide Research, Cambridge, MA, USA. [5]Spectrix Analytic Services, LLC., North Haven, CT, USA. ✉e-mail: aal44@cam.ac.uk

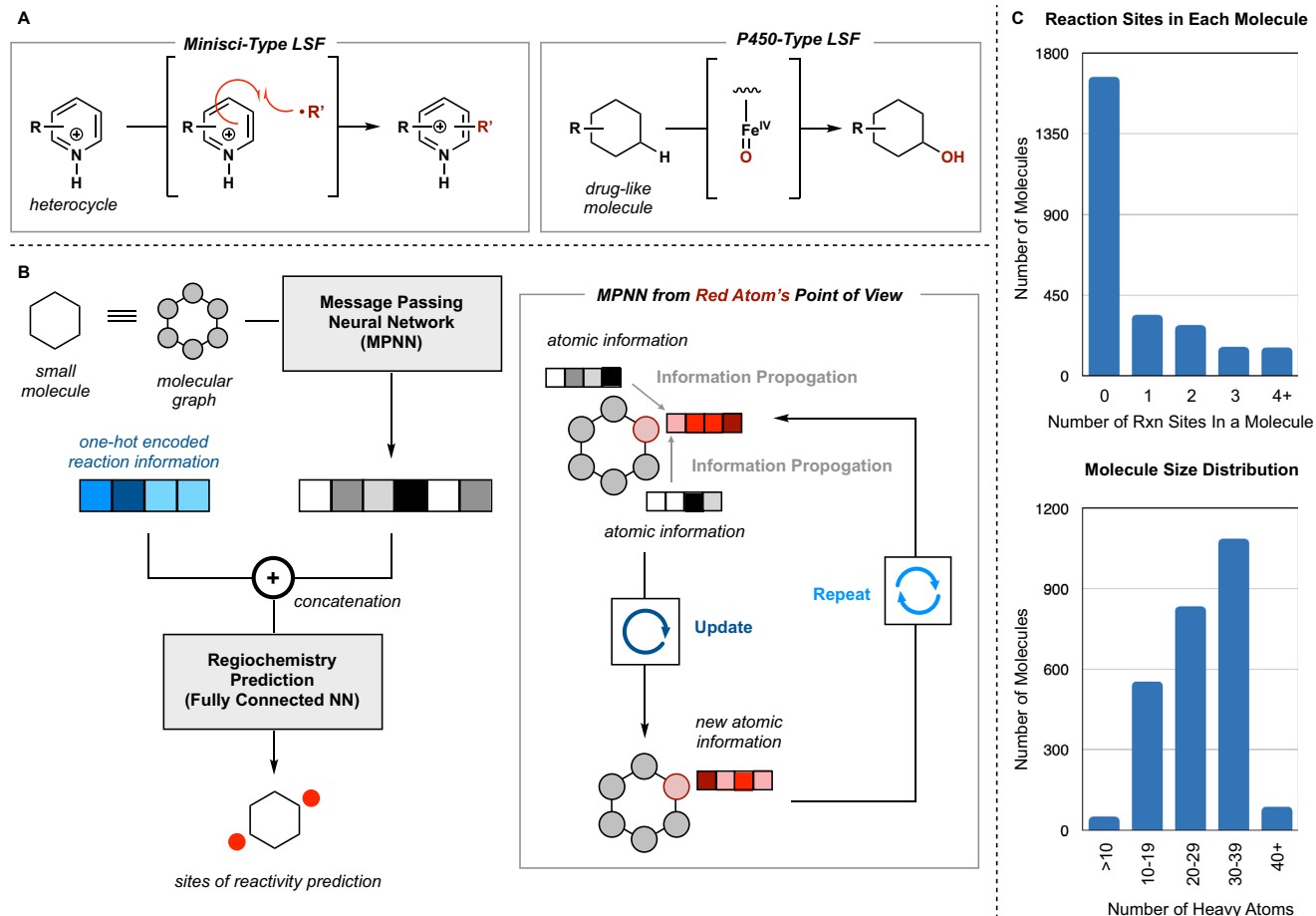

**Fig. 1 | Overview of the model framework, reactions modeled, and model dataset.** Source data for dataset breakdown is provided in the source data Excel file. LSF = late-stage functionalization, NN = neural network. **A** Mechanistic difference between the one-electron-based transformations of the two major types of reactions in the dataset: Minisci and P450. **B** Graphical overview of the basic message passing neural network (MPNN) model. Molecules are represented as graphs, to go through the MPNN, where atom information is propagated to its through-bond neighbors. The resulting embedded molecule (featurized molecule) is then concatenated with the one-hot encoded reaction information. This resulting vector is given to the final neural network to predict the probability of functionalization of each atom. **C** Distribution of reaction sites per molecule and molecule size in the dataset. The inclusion of negative data (0 reactive sites) was key to model performance. The majority of LSF molecules were between 20 and 40 heavy atoms (non-hydrogen atoms).

more plentiful and significantly less noisy than real-world data, notable performance can be achieved with carefully curated literature datasets. Some experimentally based reactivity models can reach human expert performance in their predictions[19]. However, ML-based regiochemical prediction is still difficult. Due to the challenges of characterizing the regiochemical outcomes of thousands of reactions, experimental data-based models must often operate in lower data environments, and if gathered from the literature, often with data that contains few negative data points, i.e., molecules that don't react. In contrast, datasets that include easily extractable yield information often contain ten-fold more data[26]. This makes it more difficult for ML to find relationships between the molecular structure and LSF outcomes.

Herein, we report a solution to this problem: the utilization of open-source $^{13}C$ nuclear magnetic resonance (NMR) data in conjunction with LSF data. We hypothesized that an ML model, with its high parameterization, would offer an improvement in accuracy when predicting the regiochemical outcomes of more complex molecules (Fig. 1B). Our model is a graph-based model that does not require precomputed molecular properties nor any 3D molecular information for accurate regioselectivity prediction. As a proof of concept, we highlight our framework's predictive ability on both Baran and Molander-type Minisci and P450 LSFs, transformations whose substrate scope is well defined. We show that our model outperforms the Fukui function-based index predictions, and two accurate, previously reported,

reactivity-based machine-learning models: one 1-electron-based enzymatic reactivity model and one 2-electron-based small-molecule model. This LSF predictive framework has application towards the development of rapid and facile access to a diverse array of drug-like compounds, specifically with respect to structure-activity relationship (SAR)-probing synthesis and expanding the known chemical space available for exploration.

## Results and discussion

### The dataset

Data was sourced from Pfizer's internal medicinal chemistry dataset which consisted of ~2600 reactions, 647 unique molecules, and 823 unique LSF conditions. The majority of these reaction conditions were Minisci-type functionalizations (1928 reactions), including Minisci reactions utilizing the Baran Diversinates™ (463 reactions)[27]. Classic Minisci conditions were included in the training set, however, the majority of the training data consisted of Baran and Molander Minsci reactions (Table S1). Additionally, other single-electron-based late-stage functionalizations were included in the training data such as P450 catalyzed oxidations (642 reactions), electrochemical methylations (12 reactions), and photoredox alkylations (93 reactions) (see Table S2 for further breakdown of the dataset). Reactions that yielded oxidative cleavage or hydrolyzed side products were kept. A key facet to our dataset was the inclusion of data that contained unsuccessful

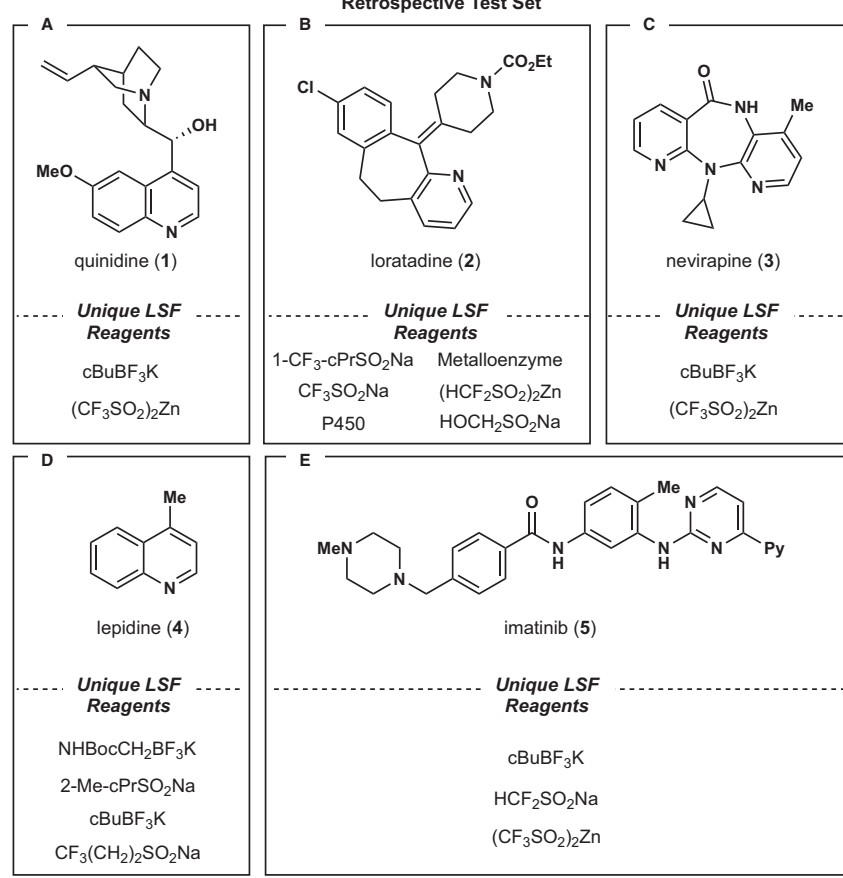

**Fig. 2 | The retrospective test set used for optimization of the models.** The conditions used for each test set molecule are shown below the corresponding structure. LSF = late stage functionalization. **A** quinidine—the standard Baran and Molander conditions, **B** loratadine—containing the widest variety of conditions indicative of its reactivity, **C** nevirapine—the standard Baran and Molander conditions, **D** lepidine—both zinc and sodium sulfinate Baran Diversinates™ are used, **E** imatinib—both zinc and sodium sulfinate Baran Diversinates™ are used.

conditions that led to no significant product formation (zero reactive sites). Despite the significant mechanistic differences between these reaction classes, we hypothesized that additional chemical information relating to the inherent reactivity of both the reagent and the molecule would be advantageous to regiochemical outcome prediction (Fig. S1). A mixture of reaction classes has seen success when utilized in other reactivity-based predictions[19,25]. To implicitly distinguish between the reaction types, each unique reagent, oxidant, solvent, additive, and acid was one-hot encoded to form a specific reaction vector, unique for each unique reaction condition. Similar to an organic chemist, the selectivity neural network (Fig. 1B) would need to interpret the mechanism type from the collection of reagents.

When deciding the correct method to split the data into training and testing sets, we opted for scaffold-based instead of a random split. It has been hypothesized that a random split encourages the model to simply memorize the inherent reactivity of a molecule, instead of applying its learned chemical knowledge to new scaffolds[28]. A scaffold split, where every molecule in the test set is an unseen molecule, provides a more challenging target. The retrospective test set consisted of 25 reactions which were comprised of 5 unique molecules and 17 unique reaction conditions. Of the reaction conditions, 22 were Minisci-type functionalizations with 4 utilizing the Baran Diversinates™, one was a P450 oxidation, and one was a metalloenzyme oxidation (Fig. 2).

## The model
One artificial intelligence architecture that has seen good performance has been message passing neural networks (MPNNs), a subset of graph convolutional neural networks (GCNNs), first utilized by Duvenaud et al., Li et al., and Gilmer et al. in the mid-2010s[29–31]. MPNNs are a robust and versatile way to predict macro properties (i.e., solubility, compound assay activity, IR spectra, energy)[30,32–34] and micro properties (i.e.,$^{13}$C and $^1$H NMR shifts, regioselectivity)[24,35] of molecules by representing molecules as graphs. Graphs, in mathematics, are structures made up by nodes and edges; nodes are concrete entities (events, people, atoms, etc.) and edges indicate that two things have a connection (these events happened due to the same cause, these people all know each other, these atoms share a bond). Briefly, MPNNs work by transmitting information from one node to another via the edge highway. Each message pass transmits the atom's information one bond further away, radially, with the intention that after a sufficient number of message passes, each atom will have a comprehensive understanding of its local environment (Fig. 1B)[30].

We developed an MPNN that sits at ~100 lines of code making it fast, easy to work with, and flexible. The implementation of the MPNN and the trained models can be found at: https://github.com/emmaking-smith/SET_LSF_CODE[36]. We believe this is the first study that discloses predictive LSF models trained on a large-scale dataset across a drug-like chemical space comprising both positive and negative results. The MPNN was designed to take in basic atomic information (atomic number, atomic symbol if the atom was a hydrogen acceptor or donor, its hybridization, if the atom was aromatic or not, and the number of explicit hydrogens) and basic structural information (the connectivity of each atom to its neighbors and the type(s) of bonds used in those connections). If the chemist did not know molecular property X by looking at the structure, that

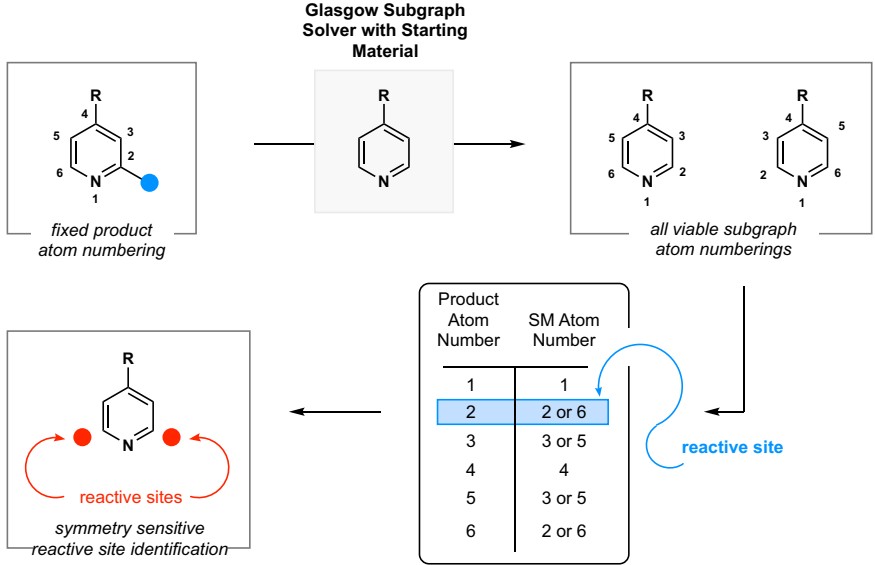

**Fig. 3 | Diagram of workflow for identification of reactive sites in symmetric molecules.** All possible starting material (SM) to product atom mappings are generated with the Glasgow Subgraph Solver (creation of atom number conversion table). The reactive site is identified via a change in carbon oxidation (blue highlight) and all corresponding SM indices are labeled as reactive. This technique preserves the symmetry of the SM atom sites.

information would not be given to the model either. Rather the model must infer relevant chemical and spatial information from the structure. From this information, the MPNN would synthesize an embedded molecule vector which would then be concatenated with the reagent-specific one-hot encoding and run through a feed-forward neural network to classify each atom within a molecule as reactive or not reactive (unreactive).

### Finding the reaction centers

The first challenge to overcome was to establish automated extraction of reactive sites, the labels for the ML task at hand. Reaction center identification is a challenging area of research[37,38] and for our regioselectivity prediction, we required the atom index(es) of the carbon atoms that changed in oxidation state. Visually, this is a trivial task, but due to the arbitrary nature of atom indices across chemoinformatics programs, it becomes much more challenging to perform this automatically. One possible solution is to use atom-mapped SMILES strings, where every atom in the product has been traced back to its corresponding atom in the starting material[39]. However, we believed a more user-friendly approach was possible. For our style of LSFs, the core structure of the molecule remained unchanged, with only the extremities exchanging a hydrogen atom for a more complex motif. Therefore, the starting materials were mathematically linked: the starting material was a *subgraph* of the product. In mathematical terms, a subgraph is a graph formed by nodes and edges that are only within its parent graph. From the molecular point of view, a subgraph could be a moiety within a molecule or the core of a molecule. The recent development of a fast, accurate, open-source Glasgow Subgraph Solver was the key to automatically finding the starting material subgraph within the product structure, facilitating the extraction of reactive sites[40]. Code for the molecule SMILES to reactive site pipeline can be found at: https://github.com/emmaking-smith/SET_LSF_CODE[36]. In addition to automating the task of finding the LSF reaction centers without the need for atom mapping, the workflow is specifically set-up to deal with symmetry in molecules. The Glasgow Subgraph Solver was directed to find all possible subgraph solutions for a given starting material and product, elucidating all possible starting material-to-product atom mappings. Upon identification of

the carbon atom indices whose oxidation state had changed, all corresponding starting material atom indices, including the symmetric indices, were identified and labeled as reactive (Fig. 3). For degradation byproducts, the fragmentation from the resulting oxidation was oftentimes too dramatic for the starting material to remain a subgraph of the product, resulting in 6% of the reactions needing manual elucidation of the reaction center.

### The loss function

With a model architecture and accurately labeled data in place, we turned our attention to the choice of the loss function, the system that penalizes the model and directs the learning. Loss functions can be broadly divided into two categories, regression or classification, where regression loss functions are used with regression tasks and vice versa[41]. Our task was to classify each atom in a molecule as a member of the reactive class or not a member of the reactive class (unreactive) thus classification loss functions were appropriate. The Binary Cross Entropy (BCE) loss, which penalizes the model based on the log-likelihood of correct class prediction, was chosen (Eq. S2). A challenge with reactivity and regioselectivity prediction is that most atoms in a given molecule are unreactive. Our most reactive molecule had only 30% of its structural atoms reacting, leaving 70% of its atoms unreactive and most molecules in our training data had 1 or fewer reactive structural atoms (Fig. 1C). Therefore, a model can be technically accurate by simply predicting that all sites are unreactive, though such model would be practically useless. What was required was a loss function that could more heavily penalize incorrect predictions and give less weight to correct unreactive predictions. To this end, a variety of BCE loss weightings were investigated, whose central theme was that the weight given to correct class predictions was inversely correlated to the frequency that that class was predicted (Eq. S2–Eq. S4); the value of each correct reactive site prediction was tempered by how often the model predicted any given atom was reactive, and vice versa for unreactive site prediction.

### Model results: retrospective test set

The baseline model was a random forest, which is known to be an excellent predictor of molecular features (e.g., compounds increasing

the lifespan of *C. elegans*, $IC_{50}$ measurement prediction of drug-like molecules, excitation energies, and associated oscillator strengths of fluorophores) especially in low-data environments[42–44]. Molecules were encoded as their atom-wise Morgan fingerprints. Each row corresponds to the Morgan fingerprint of a specific atom within the molecule. The corresponding one-hot encoded reaction vector was concatenated to the atom-wise Morgan fingerprint and a random forest classifier was then used to predict whether or not each atom in the molecule was reactive or not reactive. We used the well-established classification accuracy metric of the *F*-score, which balances precision and accuracy to judge model performance. Two other metrics, accuracy (total correct reactive sites predicted/all possible reactive sites) and area under the receiver operating curve (AUROC) are also given for additional interpretability of performance[45]. Initial results on our test set revealed a modest *F*-score of 0.42 (Accuracy = 94%, AUROC = 0.67), with Fukui-index-based predictions yielding a lower *F*-score of 0.19 (Accuracy = 90%, AUROC = 0.57) (Fig. 4A). Fukui indices are predicted only for the molecule, not for the reagent, however, distinctions between different regents are entirely possible. Nucleophilic Fukui indices, $F_i(+)$, correspond to regiochemical outcomes utilizing electrophilic radicals (•CF_3), and radical Fukui indices, $F_i(0)$, correspond to regiochemical outcomes utilizing nucleophilic radicals (•CF_2H, •cBu) (see SI pg. S5 for a mathematical description of each index)[14,15]. For any radical whose electrophilicity/nucleophilicity reactivity was uncertain, the Fukui indices that best fit the experimental reactivity were used for the calculation of the *F*-score.

Evaluation of these initial predictions suggested that the model was challenged with extended conjugated systems, such as those present in loratadine (**2**) and imatinib (**5**). We hypothesized that this was due to the difficulty of atoms in one hemisphere of the molecule seeing atoms on the other hemisphere in the MPNN. Whilst increasing the number of bonds that every atom's information travels between (the range of the atom's message) did not improve performance, the incorporation of a universal node did. This universal node, as described by Gilmer et al. (Gilmer et al. used the term master node), is an all-seeing node−information from every atom is given to the universal node, which in turn gives information to every atom about distant atoms[30]. Implementation of a universal node MPNN led to a model with a modest increase in *F*-score to 0.46 (accuracy = 94%, AUROC = 0.72) (Fig. 4A).

At this point, we suspected we were running up against the limit of the data. Ideally, this would be solved by performing additional LSF reactions, however, this data is laborious and expensive to generate. Every regioisomer must be isolated and characterized for every new substrate which can be cost and/or time prohibitive. Another obvious solution would be to increase the amount of information in each atom's featurization for a deeper understanding of chemical environments. However, given the poor performance of QM-derived atomic descriptors for MPNN regioselectivity prediction in LSF, alternative solutions were sought out first (see the Quantum Chemistry Augmentation Section for a detailed discussion)[24]. Thus, transfer learning was employed. This is a technique whereby a model is trained on off-task data before being trained on the desired-task data to boost performance[46]. It was crucial to choose a transfer learning task that had significantly more data than our current training set which would allow for more complex correlations between structure and reactivity to be inferred. However, it was also imperative that this off-task bore some relationship to atomic reactivity. We hypothesized that ¹³C NMR shift prediction would be uniquely suited for our goal, which can be abstracted as quantification of local chemical environments. In addition, the inherent symmetry of a molecule is represented in NMR spectra as atoms with identical chemical environments have identical NMR shifts[47]. This would transfer to atoms with identical chemical environments that have identical reactivity. Thus, ~27,000 open-source ¹³C NMR shifts were obtained from Jonas et al.'s previous work

(originally sourced from NMRShiftDB), and transfer learning from ¹³C NMR shift to LSF regioselectivity prediction commenced[35]. This step enabled a major improvement in model performance with the top-performing model, MPNN_LSF, yielding an *F*-score of 0.62 (accuracy = 96%, AUROC = 0.79) (for every 1 *true positive*, 1.25 incorrect sites are obtained) and an average model performance over 5 initializations of 0.57 (accuracy = 96%, AUROC = 0.75) (Fig. 4A). Interestingly, we observed that negative data was important for model performance. Removing the entries with zero reactive sites (unproductive reaction conditions) led to a substantial decrease in model performance (Fig. S4). We hypothesize that this is because the negative data allows the model to infer similarities between different one-hot encoded reaction conditions.

## Comparison to other machine-learning models

To highlight the difficult nature of predicting Minisci-type transformations without this ¹³C NMR pretraining protocol, we investigated how other graph-based architectures would perform on our retrospective test set. A recently developed neural network by Jensen et al. utilized a joint network approach for 2-electron-based regioselectivity prediction. Their first neural network predicted on-the-fly QM properties, which were then given to their second neural network that classified which product was the major product from a user-generated list of possible structures. This approach, dubbed ml-QM-GNN, saw excellent top-1 accuracy performance even in low training data regimes and was validated on a broad range of 2-electron-based transformation classes, with a top-1 accuracy of over 85%. To investigate Minisci-based transformations, we transformed our dataset into the correct format, first elucidating all possible mono-addition C-H functionalizations given our reagent, followed by complete atom mapping of each reaction[48]. Using default parameters, ml-QM-GNN was trained on our training dataset and tested against our retrospective test set. Accuracy was determined using ml-QM-GNN's criteria of top-1 accuracy, where the overall retrospective test set accuracy was the ratio of correctly predicted major products to the total number of reactions. As many reactions contained multiple correct possible products, the ml-QM-GNN's classification was deemed correct if its top-1 prediction was any of the valid possible products. Over 5 initializations, the average top-1 accuracy of ml-QM-GNN was 11%, compared to an average top-1 accuracy of 71% for our ¹³C NMR transfer learning model (Fig. 4C).

Finally, we compared our results to a graph-based model specifically developed to predict the outcomes of single-electron-based transformations: Meta-UGT[49]. Meta-UGT was developed to predict the site of metabolism of UDP-glucuronosyltransferases (UGTs). The natural promiscuity of these phase II metabolic enzymes renders reactivity prediction challenging. The model works in two phases, first predicting if a small molecule is a substrate for the enzyme, followed by the site-specific predictions. When tested upon drug-like molecules, Meta-UGT achieved a top-1 site of reactivity prediction accuracy of 89%, making it a suitable candidate to test our model against. Thus, Meta-UGT was trained with default parameters on our training data and tested on the retrospective test set, yielding an average top-1 accuracy of 42% (Fig. 4C).

## Model results−P450-only test set

To investigate this training technique's performance, we devised a different regioselectivity task: P450 oxidation. P450 oxidation plays a central role in drug metabolism, determining the efficacy and duration of a pharmaceutical. Additionally, the interactions of some drugs with human P450s are known to inhibit and/or induce P450 activity leading to drug–drug interactions[50,51]. Due to its inherent promiscuity[52,53], P450 oxidations are a promising LSF and an excellent test for our framework. Mechanistically distinct from Minisci functionalizations, the Fe(IV)-oxo complex acts upon the substrate via radical rebound or

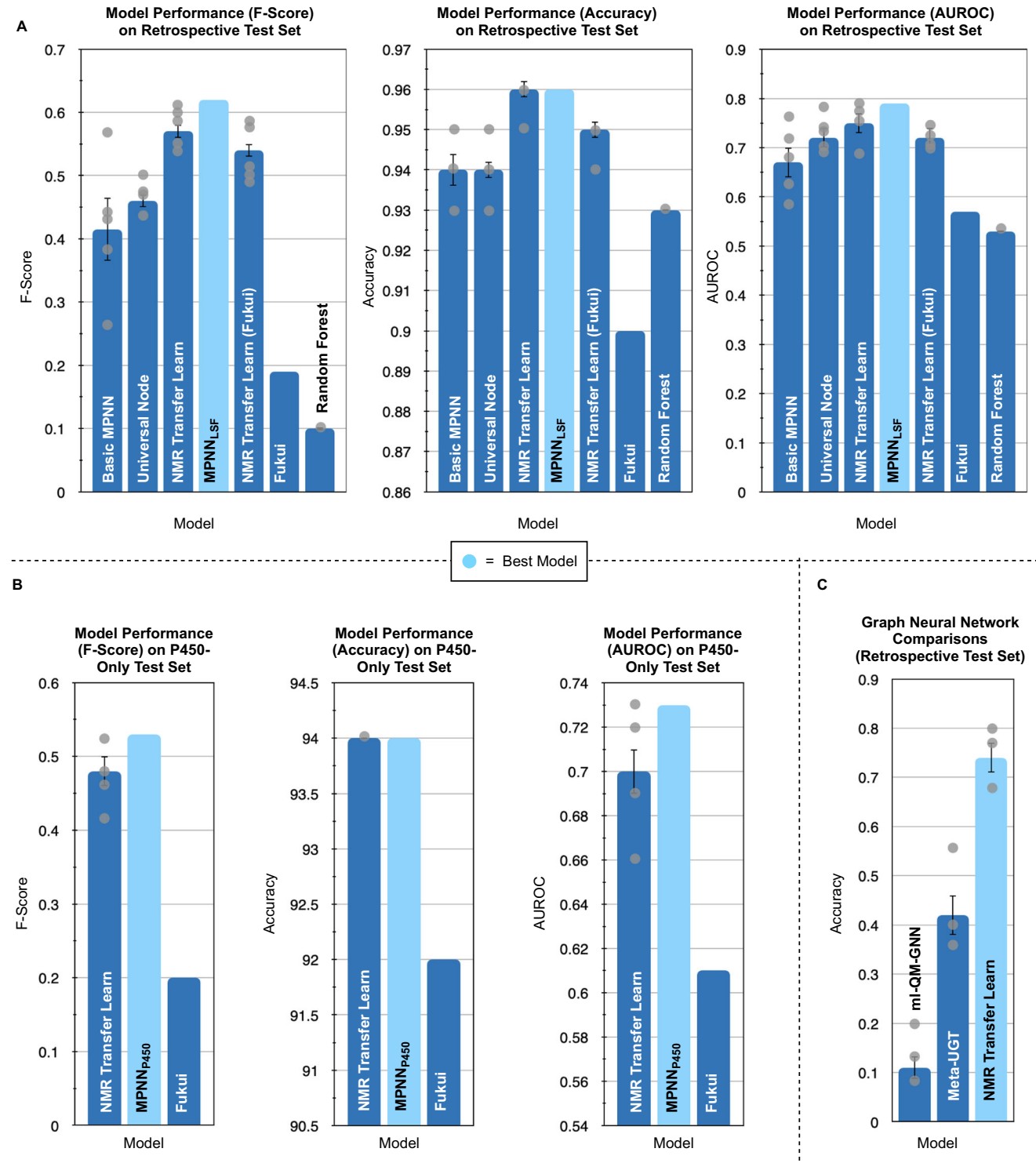

**Fig. 4 | Model performance on retrospective and P450-only test sets.** Average model performance on 5 initializations ($n = 5$) with each architecture on the test sets. A basic message-passing neural network (MPNN) is the baseline graph neural network ($n = 5$). The universal node is the MPNN architecture with the inclusion of a universal node ($n = 5$). Nuclear magnetic resonance = NMR. NMR transfer learn is the transfer learned model without Fukui-index augmentation ($n = 5$). NMR transfer learn (Fukui) is the transfer learned model with Fukui-index augmentation ($n = 5$). The best model on the retrospective test set is highlighted in light blue ($n = 1$). Fukui is prediction solely from Fukui indices ($n = 1$). Random Forest predictions are from a random forest classifier ($n = 5$). The bars in the bar charts represent the average when $n > 1$, with gray dots representing the individual data points (initializations with identical values are shown as a single point). Standard error bars are shown. Source data for each bar chart can be found in the source data Excel file. **A** Performance (F-score, accuracy, and area under the receiver operating curve (AUROC)) on the retrospective set. **B** Performance (F-score, accuracy, AUROC) on P450-only test set with $^{13}C$ NMR transfer learning. **C** Comparison of top-1 accuracy for two graph reactivity models originally developed for 2-electron-based (ml-QM-GNN) and 1-electron-based (Meta-UGT) transformations ($n = 5$ for all).

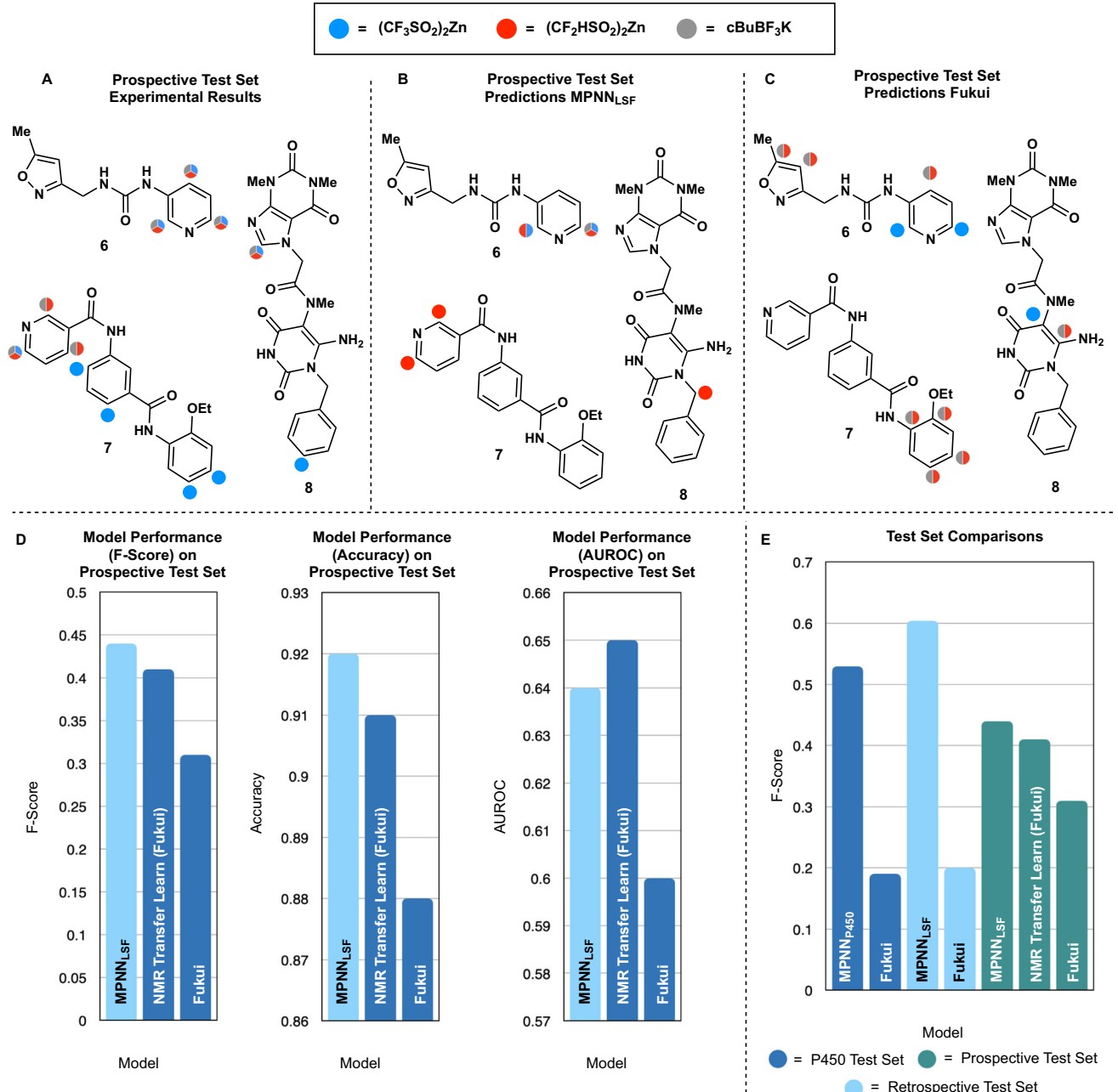

**Fig. 5 | Results on the prospective test set and best models overall.** Model performance was judged from a single run ($n = 1$) (nuclear magnetic resonance = NMR). Source data for each bar chart can be found in the source data Excel file. **A** Experimental results. Color-coded by reagent-specific reactivity. Split circles imply more than one reagent functionalized that position. **B** MPNN$_{LSF}$ (best retrospective model) predictions on the prospective test set. **C** Fukui predictions on the prospective test set. **D** F-score, accuracy, and AUROC (area under the receiver operating curve) reported for MPNN$_{LSF}$ (highlighted in light blue) NMR transfer learn (Fukui) is the transfer learned model with Fukui-index augmentation. **E** Comparison of the best models on each of the 3 test sets ($n = 1$). MPNN$_{P450}$ = Best P450-only test set model. Color coding corresponds to the test set used for evaluation.

through a concerted mechanism, to release the newly oxidized compound (Fig. 1B)[54,55]. Site of metabolism (SoM) prediction, which deduces the most likely positions for human P450 oxidation on a given compound, has seen great strides in the past two decades[56–61]. We offer this framework as a jumping-off point to develop an applicable, isoform-agnostic SoM methodology. Fukui-based indices have also been shown to be effective at determining the regiochemical outcomes of P450 oxidations and thus will be used as a baseline measure[62–64]. Thus, a P450-only test set of 31 reactions and 19 unique molecules (Fig. S6), reacting with 18 unique P450s was curated. Employing the aforementioned transfer learning technique to the P450-only test set resulted in an average F-score of 0.48

(accuracy = 94%, AUROC = 0.70) over 5 initializations. The top performing of these initializations, MPNN$_{P450}$, achieved an F-score of 0.52 (accuracy = 94%, AUROC = 0.73) (Fig. 4B). Despite only 25% of the training data containing P450 oxidations, MPNN$_{P450}$ outperformed the Fukui-index based reactivity predictions, showcasing the utility of $^{13}$C NMR transfer learning.

## Quantum chemistry augmentation
A lingering question was whether incorporating 3D information and/or quantum mechanical features as input to the graph would help model performance. Conformer generation and quantum chemistry calculations add computational overhead, which would limit this model's

applicability in practice. However, many MPNNs that utilize QM-derived information find a performance improvement. To this end, a variety of augmentations to the initial atomic features were attempted. However, neither 3D atomic coordinates generated from molecular dynamics (MD) simulations nor electronic information derived from atomic density functions improved overall performance (Fig. S5, SI pg. S4-S5). Interestingly, the addition of each atom's electrophilic, nucleophilic, and radical Fukui indices (see SI pg. S5 for a mathematical description of each index) did not see an appreciable *F*-score performance increase in either the prospective or retrospective test sets (Figs. 4A and 5E). It is possible that the Fukui indices may not provide any additional information for the MPNN. There have been numerous prior reports that indicate that MPNNs can accurately predict quantum chemical properties from basic atomic information, implying that an MPNN could extract the necessary quantum chemical information from barebones atom featurization, obviating the need for explicit pre-computation of quantum chemical properties[30,34,65]. This observation is congruent with Nippa et al. who independently and concurrently published an MPNN for LSF C-H borylation regiochemical and yield prediction[24]. They noted that similar augmentation of their atomic information with quantum mechanical features did not lead to a noticeable improvement of regioselectivity prediction and incorporation of 3D atomic coordinates only yielded a modest improvement over 2D molecular representations (scaffold splits). It is possible that the lack of improvement with 3D atomic featurization stems from the difficulty in characterizing properties of the LSF reaction transition state with descriptors that refer to an unperturbed substrate molecule.

## Prospective validation

With the success of our architecture in a variety of LSF regiochemical predictions, we turned our attention to assessing its ability in a completely unbiased setting through prospective prediction. Three maximally structurally different molecules were selected from Enamine's High-Throughput Experimentation catalogue via Butina Clustering. The three compounds were confirmed to not be present within the training or testing data and none had a Tanimoto similarity score over 0.35 with any molecule in the training/testing datasets, indicating low structural similarity between the three prospective compounds and the training/testing data. Each molecule was subjected to CF₂H-, CF₃-, and cBu- functionalization (Fig. 5A), and these experimental results were compared to the Fukui-derived indices and MPNN$_{LSF}$ predictions (Fig. 5B, C). Gratifyingly, MPNN$_{LSF}$ once again outperformed Fukui predictions (Fig. 5D), and the random forest baseline, even with a respectable performance of Fukui on this prospective test set. All of MPNN$_{LSF}$'s predictions made chemical sense, with predicted functionalizations occurring at known inherently reactive sites or probable sites of oxidation. Fukui predictions often yielded functionalizations at fully oxidized carbons, something that is rarely seen in these LSFs. This is perhaps due to the mechanistically agnostic behavior of Fukui-based predictions, which highlight the site(s) of the highest probability for nucleophilic/radical attack, regardless of whether or not those sites lead to productive pathways.

A deeper look at our prospective results sheds light on MPNN$_{LSF}$'s current utility, specifically its highly precise nature. For compound **6**, we see a generally good understanding of inherent pyridine electronics, which is naturally activated at the C2, C4, and C6 positions. However, the effect of the urea motif must be taken into account for a complete picture of regioselectivity. Per the governing heuristics, the π-donating nature of the urea would indicate increased reactivity at the C4 and C6 positions for electrophilic radicals (•CF₃) and reduced reactivity for nucleophilic radicals (•CHF₂, •cBu)[13]. Experimentally, it is revealed that the urea motif makes little impact on the electronics of the pyridine, however, MPNN$_{LSF}$ does not capture this. It instead hedges its bets, correctly finding C2 to be reactive for all three radicals

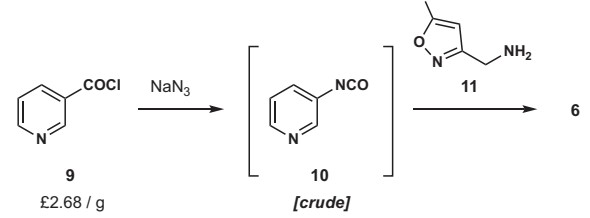

Fig. 6 | **Potential challenges of a structure-activity relationship (SAR) campaign of compound 6.** Literature synthesis of compound **6** and the cost of purchasing fluorinated starting materials for a potential SAR campaign.

but failing to predict the full chemical reactivity at C4 and C6. This may be in part due to the rarity of the urea motif within our dataset. Out of the ~2,600 training and testing molecules, only 12 contained a urea motif (~0.5% of the data), and of those 12 molecules, functionalization occurred on heterocycles distal to the urea motif. Despite this, MPNN$_{LSF}$ found 5/9 reactive sites and none of the sites it predicted to be reactive were incorrect.

For compound **7**, we once again see correct ortho reactivity for •CF₂H, however, miss the para reactivity for all radicals, perhaps owing to the more sterically congested landscape at that site. However, the clear failure of MPNN$_{LSF}$ was its inability to understand the promiscuous nature of •CF₃ functionalization on **7**. In the majority of Minisci functionalizations, the role of nucleophile is played by the radical, even for electrophilic radicals like •CF₃, and the of role electrophile is played by the heteroarene[13]. Functionalization generally occurs at a (reasonably) electron-deficient site. However, compound **7** does not completely follow this trend: all but one of the •CF₃'s functionalizations occur on non-heterocyclic, more electron-rich arenes, instead of the canonical pyridinyl motif. This atypical substitution pattern plays a large role in the lower performance of MPNN$_{LSF}$ and is even unlikely to be predicted by an expert chemist, highlighting the current limitations of our model: surprising experimental outcomes also surprise MPNN$_{LSF}$[66].

In compound **8** we finally see a small decrease in MPNN$_{LSF}$'s precision. Instead of identifying the inherently most reactive site on the imidazole, a benzylic oxidation is predicted. The predicted reactivity to difluoromethylation conditions on **8** is likely predicting the major product to be an oxidation byproduct, where the benzylic hydrogen is extracted from the generated alkyl radical and subsequently quenched via TBHP[67]. A prediction of this nature is most likely due to the decision to include byproduct reactions in the training data and lends credence to the hypothesis that the model understands general chemical reactivity trends.

From this analysis, we see that a general trend is the high precision of MPNN$_{LSF}$. This has ramifications in SAR studies, which seek to identify the best decoration of molecular scaffolds for optimal pharmacokinetic properties[68]. In a typical SAR synthesis, one motif is varied and the rest of the molecular structure is held constant. Syntheses of SAR derivates are generally convergent, with the varying motifs brought into the synthesis modularly. Despite this workflow's streamlined approach, it still requires each SAR derivative to have its own unique route. A more efficient synthesis would use one reaction to generate multiple desired products. Take compound **6** as an example,

with a known route from commercial nicotinoyl chloride (**9**) in an efficient 2-step procedure (Fig. 6)[69,70]. Aryl isocyanate **10** is formed via a Curtius rearrangement, followed by quenching with amine **11** to produce **6**. Current trends in therapeutic molecules have seen the incorporation of fluorinated functional groups as substituents on aromatic systems, such as $CF_3$ and $CF_2H$, to yield molecules with improved pharmacokinetic properties including lipophilicity, metabolic stability, and cell membrane permeability[71–73]. Indeed, approximately 20% of all approved pharmaceuticals contain some fluorine-based group[74]. An SAR campaign to investigate the effect of a trifluoromethyl at C2 and C6 would require purchasing the corresponding trifluoromethylated nicotinic acid **12**/nicotinoyl chloride **13**. However, in addition to the added cost of these starting materials (84- and 33-fold more expensive per gram, respectively), the chemist is faced with the challenging task of optimizing and characterizing the outcomes of two small-scale, multi-component, multi-step routes[75,76]. With MPNN$_{LSF}$'s precision, a chemist could be confident that a single route could provide multiple desired derivates in one fell swoop, saving the cost of starting material and most importantly, time, both in reaction optimization and in compound characterization. The lower recall isn't as problematic, as any additional bonus products can be isolated from the crude reaction mixture concurrently with the correctly predicted functionalizations. The benefit of MPNN$_{LSF}$ becomes more apparent when more exotic functional groups are investigated in SAR. Exploration of difluoromethylation at C2 and C6 by purchasing the necessary difluoromethyl starting pyridines **14** and **15** would be exceptionally expensive: 296- and 56-times more expensive per gram, respectively, of which **15** requires a carbonylation further increasing the time to derivatization[77,78]. Thus, even without perfect accuracy, MPNN$_{LSF}$ can guide SAR syntheses to produce a multitude of functionalized compounds with minimal time burden.

The regiochemical outcomes of LSF radical-based transformations are governed by many factors: the nucleophilicity of the radical, the BDE of the molecule's atoms, and the steric and electronic landscape to name a few. Interestingly, it has been observed that additional QM-derived or MD-derived data does not yield appreciable improvements in regiochemical outcome prediction. We showcase a transfer learning methodology based upon $^{13}C$ NMR shift prediction which boosts the performance of zinc sulfinate and BF$_3$K salt Minisci reaction regiochemical outcome prediction above that of the accurate Fukui-index reactivity scores, and of two reactivity prediction machine-learning models, on a narrow yet well-defined slice of chemical space. Promising predictive accuracy was also achieved on P450 enzymatic oxidations, a chemistry with a broader scope than the aforementioned Minisci conditions. Model performance was also highly contingent on the inclusion of negative data in the training set. This paradigm stands as a proof of concept for future applications in other LSF regiochemical predictions with the current best model showing potential in diversity-oriented SAR synthesis. Our $^{13}C$ NMR data is open-source and we anticipate that the incorporation of larger proprietary $^{13}C$ NMR datasets as the first step in this transfer learning methodology will expand this methodology to include in other LSF chemistry.

## Methods

### Materials

Liver microsomes were purchased from the following vendors: female mouse, male rat, male cynomolgus monkey and non-transfected microsomes (Corning, Woburn, MA); dexamethasone-induced male rat, male hamster, male dog and pooled male & female human (prepared in-house at Pfizer, Groton, CT); and male guinea pig and male rabbit (Xenotech, Lenexa, KS). Recombinant human P450 enzymes heterologously expressed in microsomes from Sf9 cells were custom-prepared by Panvera (Madison, WI).

### High-throughput biocatalytic screens

The reactions were set-up in two 96-well arrays using miniature $8 \times 20$ mm (0.2 mL) glass vials under standard glove box conditions ($H_2O$ and $O_2 < 20$ ppm). A $8 \times 20$ mm (0.2 mL) glass vial equipped with a stir bar was dispensed the reaction solvent (100 μL, 4.0 mM) followed by a solution of 1 (5.0 μL, 0.4 μmol), added as a 0.1 M solution in dichloroethane. Stirring was initiated before the metalloporphyrin (4.0 μL, 0.04 μmol) was charged, as a 10.0 mM solution in dichloroethane. The vial was treated with a 0.1 M solution of imidazole (2.4 mL, 0.24 μmol) in $H_2O$, followed by a 0.4 M solution of formic acid (4.0 μL, 0.16 μmol) in $H_2O$. Finally, the oxidant (8.0 μL, 0.08 μmol) was added as a 0.1 M solution in dichloroethane. The reaction vial was crimp sealed with a polytetrafluoroethylene (PTFE)/Silicone/PTFE septa to the glove box environment before the reaction was left to stir at 25 °C for 18 h. After this time period, the reaction was diluted with acetonitrile (0.2 mL) and analyzed directly by ultra-performance liquid chromatography-mass spectrometry (UPLC/MS). The UPLC/MS method used a 0.1% AcOH/NH$_4$CO$_2$H/$H_2O$ gradient over 0.8 min, running from 5–95% acetonitrile using a Waters Acquity UPLC BEH C18 $30 \times 2.1$ mm column at 100 °C with a flow rate of 2.5 mL/min and a detection wavelength of 210–360 nm. 0.5 μL injections were made directly from diluted reaction mixtures and ionization was monitored in positive mode.

### Baran diversinate™ late-stage functionalizations

To 1-dram pressure release vial containing Diversinate™ sulfinate reagent as sodium or zinc salts, e.g., RSO$_2$Na or (RSO$_2$)$_2$Zn (3 eq–6eq), was added to a solution of the test substrate molecule (~2 μmol, 1 eq) in dimethyl sulfoxide (DMSO) (~70–100 μL, 30 mM) and TFA (4 eq) followed by *tert*-butyl hydroperoxide, 70% in water (5 eq) at room temperature. The resulting reaction mixture was capped and heated to 50 °C overnight. The crude reaction mixture was dissolved in 3:1 acidic mobile phase (1% acetonitrile, 0.1% formic acid) and acetonitrile (~3 mL) then purified via HPLC (XSelect 5 μm C18 130 Å, $250 \times 10$ mm @ 2 mL/min). The respective fractions were pooled, and the solvent was removed using the EZ-2 Elite Genevac (3-h HPLC setting, 34 °C/238 mbar to 41 °C/7 mbar). Each isolate was characterized by MS and NMR. Due to the low amounts of isolates generated, gravimetric mass analysis is not possible; qNMR in conjunction with the enhanced sensitivity using a 1.7 mm micro-cryoprobe in DMSO-$d_6$ solvent was used to determine the concentration of the sample.

### Molander BF$_3$K salt late-stage functionalization

To 1-dram pressure release vial containing the test substrate molecule (~2 μmol, 1 eq), potassium trifluoroborate salt of the radical (1.5–2 eq), in a 1:1 mixture of acetic acid and water to make a 30 mM solution and Mn(OAc)$_3$ was added in one portion. The resulting reaction mixture was capped and heated to 50 °C overnight. The crude reaction mixture was dissolved in 3:1 acidic mobile phase (1% acetonitrile, 0.1% formic acid) and acetonitrile (~3 mL) then purified via HPLC (XSelect 5 μm C18 130 Å, $250 \times 10$ mm @ 2 mL/min). The respective fractions were pooled, and the solvent was removed using the EZ-2 Elite Genevac (3-h HPLC setting, 34 °C/238 mbar to 41 °C/7 mbar). Each isolate was characterized by MS and NMR. Due to the low amounts of isolates generated, gravimetric mass analysis is not possible; qNMR in conjunction with the enhanced sensitivity using a 1.7 mm micro-cryoprobe in DMSO-$d_6$ solvent was used to determine the concentration of the sample.

### Molecular dynamics simulations

Molecule conformations were generated with MOPAC at the PM7 level of theory[79]. The underlying molecular dynamics (MD) driver was the atomic simulation environment (ASE) package[80]. A Langevin thermostat controlled the temperature. First, the molecular geometry was

optimized followed by equilibration to 500 K for 2.5 picoseconds with a timestep of 0.25 femtoseconds. Upon equilibration, conformations were sampled every 2 picoseconds from a production run of 200 picoseconds in the NVT ensemble (constant temperature, constant volume) at 500 Kelvin, using a timestep of 0.5 femtoseconds with the same thermostat. This yielded a total of 100 configurations per molecule.

### Calculating the Fukui indices

Reactivity indices for electrophilicity and nucleophilicity for the $i$-th atom were computed by multiplying the corresponding Fukui-index [$F_i(+)$ or $F_i(\cdot)$, respectively] of the $i$-th atom by global electrophilicity/nucleophilicity for the given molecule. Fukui indices of the $i$-th atom $F_i(+)$, $F_i(\cdot)$, and $F_i(0)$ were computed as differences between the atomic charge of the $i$-th atom in the original molecule $q_i(N)$ with $N$ electrons, the charge of the same atom after adding one electron to the molecule $q_i(N+1)$, and the charge of the same atom after removing one electron from the molecule $q_i(N-1)$:

$$F_i(+) = q_i(N) - q_i(N+1)$$

$$F_i(+) = q_i(N-1) - q_i(N)$$

$$F_i(0) = \frac{q_i(N-1) - q_i(N+1)}{2}$$

For electrophilicity and radical indices, quantum chemical computations were run with PBE/6-311G, and for nucleophilicity with B3LYP/6-311G**. As partial atomic charges, Mulliken charges were used. These DFT functionals, basis sets and types of atomic charges were chosen by optimizing the predicting performance of the reactivity indices in $S_NAr$ and EAS reactions of an internal dataset of small organic molecules (unpublished). Quantum chemical computations were run in Terachem[81].

### Equations

TP = true positive, FP = false positive, FN = false negative.
$w$ = weighting value, $x$ = predicted value, $y$ = true value (always 0 or 1).

The variables, $pred_p$ and $true_p$ refer to the ratio of predicted positives to all reactive sites. A $pred_p$ of 1 indicates a model that predicts all sites react and a $pred_p$ of 0 indicates a model where every molecule is unreactive.

$$F \text{ score} = \frac{2 \cdot TP}{2 \cdot TP + FP + FN}$$

$$\text{BCE Loss} = \sum_{i=0}^{n} w_i \cdot (y_i \cdot \log(x_i) + (1 - y_i) \cdot \log(1 - x_i))$$

$$\begin{aligned}\text{BCE weight 1} = &\, x \cdot y \cdot \log(pred_p) + (1 - y) \cdot (1 - x) \cdot \log(1 - pred_p) \\ &+ (1 - y) \cdot x \cdot \log(true_p) + y \cdot (1 - x) \cdot \log(1 - true_p)\end{aligned}$$

$$\begin{aligned}\text{BCE weight 2} = &\, \Big[ x \cdot y \cdot \log(pred_p) + (1 - y) \cdot (1 - x) \cdot \log(1 - pred_p) \\ &+ (1 - y) \cdot x \cdot \log(true_p) + y \cdot (1 - x) \cdot \log(1 - true_p) \Big] \\ &+ \Big[ y \cdot x \cdot \log(true_p) + (1 - y) \cdot (1 - x) \cdot \log(1 - true_p) \\ &+ (1 - y) \cdot x \cdot \log(1 - pred_p) + y \cdot x \cdot \log(pred_p) \Big]\end{aligned}$$

$$\begin{aligned}\text{BCE weight 3} = 2 &\Big[ x \cdot y \cdot \log(pred_p) + (1 - y) \cdot (1 - x) \cdot \log(1 - pred_p) \\ &+ (1 - y) \cdot x \cdot \log(true_p) + y \cdot (1 - x) \cdot \log(1 - true_p) \Big] \\ &+ \Big[ y \cdot x \cdot \log(true_p) + (1 - y) \cdot (1 - x) \cdot \log(1 - true_p) \\ &+ (1 - y) \cdot x \cdot \log(1 - pred_p) + y \cdot x \cdot \log(pred_p) \Big]\end{aligned}$$

## Data availability

The data generated in this study have been deposited in our GitHub repository https://github.com/emmaking-smith/SET_LSF_CODE (https://doi.org/10.5281/zenodo.8252537)[36]. The full dataset is available under restricted access due to proprietary structures being present in the data. Access can be obtained by entering a collaboration or legal agreement with Pfizer and requesting permission to pfizer_LSF_NatureCommunications_14_August_2023. A literature-only dataset[45,82] of non-proprietary compounds is available at https://github.com/emmaking-smith/SET_LSF_CODE. Bar chart data are provided in the Source Data file. Source data are provided in this paper.

## Code availability

The entirety of our code, including the trained models, can be accessed at https://github.com/emmaking-smith/SET_LSF_CODE (https://doi.org/10.5281/zenodo.8252537)[36].

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

## Acknowledgements

Financial support for this work was generously provided by Pfizer and the Royal Society (Newton International Fellowship to E.K.S. and University Research Fellowship to A.A.L.). We wish to thank Rokas Elijošius, William McCorkindale, and Oliver P. King-Smith for their enlightening discussions. We are grateful to Hans Renata and Roger M. Howard for assistance in manuscript preparation. The authors would like to acknowledge several Pfizer colleagues and Spectrix vendor partners who have contributed to this work including Manjinder Lall, Gregory Walker, R. Scott Obach, and Douglas Spracklin for their leadership and execution of the Lead Diversification Platform (LDP), and Danial Morris for LDP product generation, isolations, bioanalytical support, and anyone else who has contributed to the LDP from the date of its inception.

## Author contributions

This work was conceived by E.K.S. and A.A.L. Model development and code were written by E.K.S. Experimental results were performed and collected by U.R. and B.L. and D.H. F.A.F. implemented the molecular dynamics simulations. DFT simulations were performed by A.V.S. and Q.Y. E.K.S. wrote the manuscript and all members contributed to its editing.

## Competing interests

A.A.L. is a co-founder and owns equity in PostEra Inc. and Byterat Ltd. U.R., A.V.S., and Q.Y. are employed by Pfizer Inc. B.L. and D.H. are employed by Spectrix Analytic Services, LLC. The remaining authors declare no competing interests.
