## [Peer Review File · Nature Communications]

Predictive Minisci Late Stage Functionalization with Transfer LearningREVIEWER COMMENTS

Reviewer #1 (Remarks to the Author):

This manuscript by King-Smith et al. describes an ML model for site selectivity in radical-based late-stage functionalization. The reactions described in the manuscript are of significant interest in organic synthesis, the work is well done (though see below for some suggestions for improvement) and well presented. A particular strength of the paper is that some of the predictions were tested experimentally and found to reasonably agree with the predictions.

Nevertheless, I hesitate to recommend the manuscript for publication because the work described fails in a fundamental aspect of science: it is not reproducible. The authors do a good job in making the pipeline, modules and trained models available via github, but in the absence of the underlying dataset, none of it is reproducible. This reviewer is painfully aware that (with very few exceptions), this is common practice in the field of ML applications in chemistry, but that is not a reason why the standards in this field should be lower than e.g. in computational chemistry or synthetic chemistry where documentation of the results is mandatory. This reviewer would argue that the opposite is the case in ML application where any trustworthy model needs to be reproducible, which in turn requires that the underlying data is disclosed. In the end, this might be a policy decision that needs to be made by the editor, but as a reviewer I have to insist on it.

Should the editor decide that this manuscript is publishable without the data, I would recommend publication after revisions as follows:

- At several times, the Fukui indices are presented as the state of the art (line 22) and “the model to beat” (line 232). This is in the opinion of this review a strawman. The key slide of hands is in lines 36-41 where the Ref8 is cited as showing the dependence on a range of factors including “the nucleophilicity...[] ...and the electronics of the heteroarene”, i.e. things encoded in the Fukui indices, which is then reduced to “correlate well with the indices from Fukui functions” (no reference or information of what “well means” given). It is not surprising at all that a multi-parameter NN is able to outperform a single-parameter correlation but it is not a fair comparison as other useful aspects from the Fukui function such as interpretability are lost. Therefore, it is not a “notable” result that the “Fukui indices will fail to discern differences in the solvent environment”, it is a consequence of how the authors set up the comparison. Similarly the Fukui-based indices are not “the model to beat” but rather the much more sophisticated correlation models that were build for P450-type reaction. I am sure Pfizer has several of these in-house and recommend that they are used as a more reasonable comparison.

- In line 50/51, the author make a claim [“Whilst real-world data is often noisier and not as readily available as computational DFT data, the utility of ML models built upon experimental data usually outweighs the downsides “] which is very broad and thus almost certainly wrong as anyone who tried to build a yield model on the USPTO (and there have been many attempts, see e.g. YieldBERT by Schwaller) can attest to. It always comes down to what question and what data. Thus, I suggest the authors take a more nuanced view rather than making broad pronouncements.

- The authors use the F-score as a metric for success. While this is not unreasonable for the classifier approach used here, it does have some drawback that it is not clear to a synthetic chemist who uses the model what the likelihood for a correct prediction (coming back to the “trustworthiness” above, without which the models will not be accepted) is. My suggestion would be to complement the F-score with one or two metrics that are more intuitive to chemists, for suggestions see e.g. Bender et al Nat Rev Chem 2022.

- It should be noted that the finding of the reaction centers (as the authors mention, a topic where many different approaches have been used) using a subgraph approach is specific for the LSF discussed here but will be less useful in other reactions. It would also be good to be more quantitative here, i.e. how many or what percentage of reactions needed to be handled manually

-

Reviewer #2 (Remarks to the Author):

The manuscript entitled "Predictive Minisci and P450 Late Stage Functionalization with Transfer Learning" describes the application of graph neural networks for regioselective and in general reactivity prediction for Minisci alkylations and P450 oxidations of drug-like molecules.

The presented study is very interdisciplinary, combining aspects of drug discovery, machine learning, and computational chemistry. The code is open source and the manuscript is written on a high level, making it suitable for the broad audience of Nature Communications. The introduced method is validated in retrospective and prospective experiments. Moreover, the authors discuss timely and relevant aspects for the reaction prediction community, such as the relevance of 3D information, quantum chemical augmentation, and transfer learning.

The results presented are timely and suitable for Nature Communications.

However, the following points should be addressed before the manuscript is accepted for publication in Nature Communications.

Data Set:

1. The data set consisting of 2,613 reactions, 647 unique molecules, and 823 unique LSF conditions would be of the highest interest to the community. It is not clear if this data is published. A statement regarding data availability should be added to the main manuscript.
2. It would be suitable to add the individual numbers of reactions in the data set for the mentioned additional reaction types next to Minisci-type reactions: (i) P450 catalyzed oxidations, (ii) electrochemical methylations, and (iii) photoredox alkylations.
3. It would be valuable to discuss/add the numbers of negative reactions per reaction type (maybe through a table or barplot in the SI?).
4. The hypothesis that additional reaction types, such as P450 catalyzed oxidations, electrochemical methylations, and photoredox alkylations are helpful for the overall ML task is very suitable. However, to support this statement it would be beneficial to compare the results obtained from ML models (e.g., via F-scores) trained without the data for additional reaction types to the models which include these additional reaction types in the trained set. These results could be added to the SI?
5. In my opinion Table S1 generates more confusion than clarity. It would be important to precisely show, which reaction conditions/chemicals were used for the different chemistries and also

including the number of reactions from the chemistry to get an overview of the dataset at a first glance. This information could also be included in the main.

6. The limited chemistry scope of Minisci reactions should be pointed out. Classic Minisci reactions offer a broad variety of alkylation reactions. Examples mostly show Baran-type Zn-salt and Molander BF₃K salts chemistries even though it is claimed that Minisci reactions in general have been used for training and reaction prediction. Was there additional Minisci-type reaction data used? If so, where is this data? If the data set indeed mainly covers Baran-type Zn-salt and Molander BF₃K, the limited applicability should be mentioned.

The Model

7. MPNN_{P450} is not explained in the main text, and only mentioned in Figure captions. It would help the reader if the difference of MPNN_{P450} to MPNN_{LSF} is somewhere explained in the main text - or in the SI, but I would suggest mentioning it somewhere outside of a Figure caption.

Finding the Reaction Centers

8. The introduced subgraph approach is a very elegant procedure to index reactive centers of starting materials in an automated way. Could the authors more clearly describe how they dealt with symmetry, e.g. a Minisci-type alkylation of a para-substituted pyridine (c1nccc(*)c1) in ortho position. Was only one ortho carbon labeled as reactive and the other an unreactive, or were both labeled with the same value (e.g. 0.5, 1)? And how were symmetric reactive centers identified? If symmetry was not covered by the model this could be added as an outlook or further extension.

Model Results

9. Morgan fingerprint, which is a molecular descriptor, was used as a descriptor for a random forest baseline model. It is not obvious to understand how a molecular descriptor is used to predict atomic properties. Could the authors either briefly discuss how Morgan fingerprints were leveraged as atomistic representation and how this was incorporated, or reference a paper where this was already done?

10. It is discussed that 3D atomic coordinates generated from molecular dynamics (MD) simulations did not improve the model performance. The authors could more clearly discuss how 3D information was incorporated into the model architecture. Were interatomic distances included as edge features? What cut-off radius per atom was used for atoms to be connected via edges? Were angles included? How many 3D conformations were used per molecule for model training? What software was used for the MD simulation and conformer generation? Which coordinates were added to the universal node, if a universal node was included?

Experimental, SI

11. Compound characterization for P450 and Metallozyme reactions missing in SI (including NMRs).

This characterization should be added.

12. High-resolution mass spectrometry (HRMS) data is missing. HRMS data should be added for isolated, novel, and fully-characterized products.

Comments on the chosen title:

13. Title is partially misleading. The authors claim to predict Minisci and P450 LSF reactivity. The dataset mostly contains Minisci reactions (and two P450/metalloenzyme reactions of 25 on one drug molecule (Loratadine) in the retrospective test set) - not sufficient to show that the model can accurately predict P450 and metalloenzyme reactions on drug or late-stage molecules. If the inclusion of P450 had mainly the purpose of increasing the performance on Minisci alkylations, it should be treated like this.

Reviewer #3 (Remarks to the Author):

The authors present a tool to predict regioselectivity in late-stage C-H functionalization, specifically for radical addition to heterocycles in Minisci reactions.

Before accepting to review, I had indicated that I am not an expert in the tools that are the subject of this manuscript but rather an expert in late stage functionalization method development.

I agree with the authors that there is a current lack of predictability of regioselectivity in the Minisci and related reactions. I would encourage the authors to limit their claims to such reactions, and not generalize to late-stage functionalization in general as several of the claims are only appropriate for the subset of reactions that is discussed here. There are other examples of late-stage functionalization reactions that have other limitations. This is a quick but important fix for the manuscript.

I also concur with the authors that the reaction class that they have selected is an important one. While I cannot comment on the innovation and novelty of the tools developed, I would comment that, even if successful, they are currently only described for one single subclass of a well-defined reaction, which limits the potential scope and utility. This concern may also be rather readily addressable, either by expanding conceptually or by clearly indicating the limitation of scope described here.

With respect to late-stage functionalization chemistry, no conceptual advance is presented here but then I also do not think that that is what is claimed and what the paper is about.

Given that the crux of the manuscript is beyond my expertise, I cannot make a recommendation in regards to suitability for the journal but would encourage the authors to consider my comments irrespective of future outlet.

We thank each reviewer for their feedback. Our point-by-point responses are in **bold**, with relevant manuscript excerpts in **red**.

Reviewer #1 (Remarks to the Author):

This manuscript by King-Smith et al. describes an ML model for site selectivity in radical-based late-stage functionalization. The reactions described in the manuscript are of significant interest in organic synthesis, the work is well done (though see below for some suggestions for improvement) and well presented. A particular strength of the paper is that some of the predictions were tested experimentally and found to reasonably agree with the predictions. Nevertheless, I hesitate to recommend the manuscript for publication because the work described fails in a fundamental aspect of science: it is not reproducible. The authors do a good job in making the pipeline, modules and trained models available via github, but in the absence of the underlying dataset, none of it is reproducible. This reviewer is painfully aware that (with very few exceptions), this is common practice in the field of ML applications in chemistry, but that is not a reason why the standards in this field should be lower than e.g. in computational chemistry or synthetic chemistry where documentation of the results is mandatory. This reviewer would argue that the opposite is the case in ML application where any trustworthy model needs to be reproducible, which in turn requires that the underlying data is disclosed. In the end, this might be a policy decision that needs to be made by the editor, but as a reviewer I have to insist on it.

Thank you for your kind remarks on our work. We appreciate the need for openness in ML research. Although we are unable at this time to openly publish the full dataset, we have made available a dataset of Minisci and other late stage functionalization reaction curated from peer reviewed literature. Furthermore, Pfizer has also agreed to register and index the dataset so the entire dataset can be accessed by entering an agreement with Pfizer. We have added that information to the Data Accessibility Statement.

Data Availability Statement

All code is available under the MIT License at https://github.com/emmaking-smith/SET_LSF_CODE. The repository includes a literature-only dataset of non-proprietary compounds and reactions which enables to reader to replicate our workflow (A. Bender et al. Nat. Rev. Chem. 2022, 6, 428. & N. Artrith et al. Nat. Chem. 2021, 13, 505.). The full dataset is registered with Pfizer as "pfizer_LSF_NatureCommunications_PublicationDate" which can be accessed upon entering a collaboration or legal agreement with Pfizer.

Should the editor decide that this manuscript is publishable without the data, I would recommend publication after revisions as follows:

Dr Mondelo-Martell has said in correspondence that the above steps would be sufficient.

- At several times, the Fukui indices are presented as the state of the art (line 22) and “the model to beat” (line232). This is in the opinion of this review a strawman. The key slide of hands in in lines 36-41 where the Ref8 is cited as showing the dependence on a range of factors including “the nucleophilicity...[] ...and the electronics of the heteroarene”, i.e. things encoded in the Fukui indices, which is then reduced to “correlate well with the indices from Fukui functions” (no reference or information of what “well means” given).

Thank you for your comments. We have added additional information on the Fukui indices' performance in the literature to highlight its known accuracy and have removed all references to Fukui indices as "state-of-the-art", opting for a more nuanced view of these predictive parameters.

In the literature, Fukui-based reactivity indices predict the most reactive sites of Minisci functionalization with an average accuracy of 93% (average F-score of 0.77), albeit usually on smaller, minimally functionalized molecules (Kuttruff, M. et al. ChemMedChem 2018, 13, 983. & Y. Ma et al. RSC Advances 2014, 4, 17262). We hypothesized that an ML model, with its high parameterization, can offer an improvement in accuracy when predicting the regiochemical outcomes of more complex molecules (Figure 1B).

It is not surprising at all that a multi-parameter NN is able to outperform a single-parameter correlation but it is not a fair comparison as other useful aspects from the Fukui function such as interpretability are lost. Therefore, it is not a "notable" result that the "Fukui indices will fail to discern differences in the solvent environment", it is a consequence of how the authors set up the comparison. Similarly the Fukui-based indices are not "the model to beat" but rather the much more sophisticated correlation models that were build for P450-type reaction. I am sure Pfizer has several of these in-house and recommend that they are used as a more reasonable comparison.

Whilst Fukui indices are what have been used by Pfizer to predict outcomes of Minisci reactions, we understand your concerns. We have thus added the comparison of our model to Meta-UGT, a model that predicts the outcomes of enzymatic oxidations (M. Huang, et al. Journal of Cheminformatics 2022, 14, 46), which was less accurate than our model (42% accuracy vs 71% accuracy).

Finally, we compared our results to a graph-based model specifically developed to predict the outcomes of single-electron-based transformations: Meta-UGT (M. Huang, et al. Journal of Cheminformatics 2022, 14, 46). Meta-UGT was developed to predict the site of metabolism of UDP-glucuronosyltransferases (UGTs). The natural promiscuity of these phase II metabolic enzymes renders reactivity prediction challenging. The model works in two phases, first predicting if a small molecule is a substrate for the enzyme, followed by the site-specific predictions. When tested upon drug-like molecules, Meta-UGT achieved top-1 site of reactivity prediction accuracy of 89%, making it a suitable candidate to test our model against. Thus, Meta-UGT was trained with default parameters on our training data and tested on the retrospective test set, yielding an average top-1 accuracy of 42% (Figure 4C).

- In line 50/51, the author make a claim ["Whilst real-world data is often noisier and not as readily available as computational DFT data, the utility of ML models built upon experimental data usually outweighs the downsides "] which is very broad and thus almost certainly wrong as anyone who tried to build a yield model on the USPTO (and there have been many attempts, see e.g. YieldBERT by Schwaller) can attest to. It always comes down to what question and what data. Thus, I suggest the authors take a more nuanced view rather than making broad pronouncements.

We have altered this sentence to be a less sweeping statement.

Whilst computational data is more plentiful and significantly less noisy than real-world data, notable performance can be achieved with carefully curated literature datasets.

- The authors use the F-score as a metric for success. While this is not unreasonable for the classifier approach used here, it does have some drawback that it is not clear to a synthetic chemist who uses the model what the likelihood for a correct prediction (coming back to the "trustworthiness" above, without which the models will not be accepted) is. My suggestion would be to complement the F-score with one or two metrics that are more intuitive to chemists, for suggestions see e.g. Bender et al Nat Rev Chem 2022.

Thank you for your suggestion. From your cited review (A. Bender et al. Nat. Rev. Chem. 2022, 6, 428), we chose accuracy and area under the receiver operating curve (AUROC) as two additional metrics for additional clarity on model performance and trustworthiness. These have been added to the manuscript in after each F-score in parentheses and in Figures 4 and 5D.

- It should be noted that the finding of the reaction centers (as the authors mention, a topic where many different approaches have been used) using a subgraph approach is specific for the LSF discussed here but will be less useful in other reactions. It would also be good to be more quantitative here, i.e. how many or what percentage of reactions needed to be handled manually

We have made it explicitly clear that the subgraph approach is specific for late stage functionalizations and have added the percentage of reactions we handled manually.

For our style of LSFs, the core structure of the molecule remained unchanged, with only the "extremities" exchanging a hydrogen atom for a more complex motif.

For degradation byproducts, the fragmentation from the resulting oxidation was oftentimes too dramatic for the starting material to remain a subgraph of the product, resulting in 6% of the reactions needing manual elucidation of reaction center.

Reviewer #2 (Remarks to the Author):

The manuscript entitled "Predictive Minisci and P450 Late Stage Functionalization with Transfer Learning" describes the application of graph neural networks for regioselective and in general reactivity prediction for Minisci alkylations and P450 oxidations of drug-like molecules.

The presented study is very interdisciplinary, combining aspects of drug discovery, machine learning, and computational chemistry. The code is open source and the manuscript is written on a high level, making it suitable for the broad audience of Nature Communications. The introduced method is validated in retrospective and prospective experiments. Moreover, the authors discuss timely and relevant aspects for the reaction prediction community, such as the relevance of 3D information, quantum chemical augmentation, and transfer learning.

We thank the reviewer for their positive assessment of our work.

The results presented are timely and suitable for Nature Communications.

However, the following points should be addressed before the manuscript is accepted for publication in Nature Communications.

Data Set:

1. The data set consisting of 2,613 reactions, 647 unique molecules, and 823 unique LSF conditions would be of the highest interest to the community. It is not clear if this data is published. A statement regarding data availability should be added to the main manuscript.

Thank you for catching that - a data availability statement has now been added.

Data Availability Statement

All code is available under the MIT License at https://github.com/emmaking-smith/SET_LSF_CODE. The repository includes a literature-only dataset of non-proprietary compounds and reactions which enables to reader to replicate our workflow (A. Bender et al. Nat. Rev. Chem. 2022, 6, 428. & N. Artrith et al. Nat. Chem. 2021, 13, 505.). The full dataset is registered with Pfizer as "pfizer_LSF_NatureCommunications_PublicationDate" which can be accessed upon entering a collaboration or legal agreement with Pfizer.

2. It would be suitable to add the individual numbers of reactions in the data set for the mentioned additional reaction types next to Minisci-type reactions: (i) P450 catalyzed oxidations, (ii) electrochemical methylations, and (iii) photoredox alkylations.

We have added these indications in parentheses in the main text.

The majority of these reaction conditions were Minisci-type functionalizations (1928 reactions), including Minisci reactions utilizing the Baran Diversinates™ (463 reactions).^[13] Classic Minisci conditions were included in the training set, however, the majority of the training data consisted of Baran and Molander Minisci reactions (Table S1). Additionally, other single electron based late stage functionalizations were included in the training data such as P450 catalyzed oxidations (642 reactions), electrochemical methylations (12 reactions), and photoredox alkylations (93 reactions)

3. It would be valuable to discuss/add the numbers of negative reactions per reaction type (maybe through a table or barplot in the SI?).

Thank you for the suggestion. We have added a table in the SI (Table S2) which breaks down the dataset by the number of reactions in each reaction type and the number of negative reactions.

Table S2: Dataset Breakdown

	Minisci	Zinc sulfinate Minisci	P450	Electrochem	Photoredox
Number of Total Reactions	1928	463	642	12	93
Number of negative reactions	1056	155	74	0	34

4. The hypothesis that additional reaction types, such as P450 catalyzed oxidations, electrochemical methylations, and photoredox alkylations are helpful for the overall ML task is very suitable. However, to support this statement it would be beneficial to compare the results obtained from ML models (e.g., via F-scores) trained without the data for additional reaction types to the models which include these additional reaction types in the trained set. These results could be added to the SI?

We have added a graph comparing of F-scores of the model without the removal of any data, and the removal of each reaction type (Figure S1).

Figure S1: The deleterious effect of removing various reaction types from the training data.

5. In my opinion Table S1 generates more confusion than clarity. It would be important to precisely show, which reaction conditions/chemicals were used for the different chemistries and also including the number of reactions from the chemistry to get an overview of the dataset at a first glance. This information could also be included in the main.

In retrospect, we agree. We have restructured the table to include the breakdown of reagents by reaction class, along with the number of times each condition was used in the dataset.

Table S1: Reagents Used for Each Reaction Type

Reaction Type	Reagent	Count
Minisci*	(CF ₃ SO ₂) ₂ Zn	233
	(HCF ₂ SO ₂) ₂ Zn	137
	cBuBF ₃ K	50
	Selectfluor	39
	CF ₃ SO ₂ Na	28
	CH ₃ CF ₂ SO ₂ Na	22
	HOCH ₂ SO ₂ Na	20
	MeOCH ₂ BF ₃ K	18
	(iPrSO ₂) ₂ Zn	16
	iPrBF ₃ K	15
(CF ₃ CH ₂ SO ₂) ₂ Zn	10	
Photoredox*	NFSI	31
	O ₂	26
	tBu peracetate	18
Electrochem	H ₂ O	6
	CF ₃ SO ₂ Na	5
	CH ₃ CF ₂ SO ₂ Na	1

* = Reagents used more than 10 times shown

6. The limited chemistry scope of Minisci reactions should be pointed out. Classic Minisci reactions offer a broad variety of alkylation reactions. Examples mostly show Baran-type Zn-salt and Molander BF₃K salts chemistries even though it is claimed that Minisci reactions in general have been used for training and reaction prediction. Was there additional Minisci-type reaction data used? If so, where is this data? If the data set indeed mainly covers Baran-type Zn-salt and Molander BF₃K, the limited applicability should be mentioned.

The majority of the dataset were Baran-type and Molander BF₃K, thus the test set contained those Minisci variations. We have added additional clarifications on the limited substrate scope of these reactions.

As a proof of concept, we highlight our framework's predictive ability on both Baran and Molander-type Minisci and P450 LSFs, transformations whose substrate scope is well defined.

We showcase a transfer learning methodology on ¹³C NMR shift prediction which boosts zinc sulfinate and BF₃K salt Minisci regiochemical outcome prediction performance above the accurate Fukui-index reactivity scores, and two reactivity prediction machine learning models, on a narrow yet well-defined slice of chemical space. Promising predictive accuracy was also

observed on P450 enzymatic oxidations, a chemistry with a broader scope than the aforementioned Minisci conditions, was also achieved.

The Model

7. MPNN_{P450} is not explained in the main text, and only mentioned in Figure captions. It would help the reader if the difference of MPNN_{P450} to MPNN_{LSF} is somewhere explained in the main text - or in the SI, but I would suggest mentioning it somewhere outside of a Figure caption.

We apologize for the oversight. MPNN_{P450} and MPNN_{LSF} are now both defined in the main text.

Employing the aforementioned transfer learning technique to the P450-only test set resulted in an average F-score of 0.48 (Accuracy = 94%, AUROC = 0.70) over 5 initializations. The top performing of these initializations, MPNN_{P450}, achieved an F-score of 0.52 (Accuracy = 94%, AUROC = 0.73) (Figure 4B).

Finding the Reaction Centers

8. The introduced subgraph approach is a very elegant procedure to index reactive centers of starting materials in an automated way. Could the authors more clearly describe how they dealt with symmetry, e.g. a Minisci-type alkylation of a para-substituted pyridine (c1nccc(*)c1) in ortho position. Was only one ortho carbon labeled as reactive and the other an unreactive, or were both labeled with the same value (e.g. 0.5, 1)? And how were symmetric reactive centers identified? If symmetry was not covered by the model this could be added as an outlook or further extension.

We thank the reviewer for their positive assessment of our subgraph solver solution. Yes, symmetry was taken into account by the subgraph solver, specifically by elaborating all possible accurate starting material atom numberings via the subgraph solver. In the example of a para-substituted pyridine reacting at the ortho position, both ortho carbons would be labeled as reactive as they are chemically identical in the starting material. We have expanded this section of the manuscript and added a figure for clarity.

In addition to automating the task of finding the LSF reaction centers without the need for atom mapping, the workflow is specifically set up to deal with symmetry in molecules. The Glasgow Subgraph Solver was directed to find all possible subgraph solutions for a given starting material and product, elucidating all possible starting material to product atom mappings. Upon identification of the carbon atom indices whose oxidation state had changed, all corresponding starting material atom indices, including the symmetric indices, were identified labeled as reactive (Figure 3).

Model Results

9. Morgan fingerprint, which is a molecular descriptor, was used as a descriptor for a random forest baseline model. It is not obvious to understand how a molecular descriptor is used to predict atomic properties. Could the authors either briefly discuss how Morgan fingerprints

were leveraged as atomistic representation and how this was incorporated, or reference a paper where this was already done?

We used atom-wise Morgan fingerprints to classify each atom in the molecule as reactive or unreactive. We have this added additional information in the main text.

Molecules were encoded as their atom-wise Morgan fingerprints. Each row corresponded to the Morgan fingerprint of a specific atom within the molecule. The corresponding one-hot encoded reaction vector was concatenated to the atom-wise Morgan fingerprint and a random forest classifier was then used to predict whether or not each atom in the molecule was reactive or not reactive.

10. It is discussed that 3D atomic coordinates generated from molecular dynamics (MD) simulations did not improve the model performance. The authors could more clearly discuss how 3D information was incorporated into the model architecture. Were interatomic distances included as edge features? What cut-off radius per atom was used for atoms to be connected via edges? Were angles included? How many 3D conformations were used per molecule for model training? What software was used for the MD simulation and conformer generation? Which coordinates were added to the universal node, if a universal node was included?

In the SI, we have added a section that details the molecular dynamics simulations and their incorporation into the message passing neural network.

Molecule Dynamics Simulations:

Molecule conformations were generated with MOPAC (<http://openmopac.net/>) at the PM7 level of theory. The underlying molecular dynamics (MD) driver was the Atomic Simulation Environment (ASE) package (Hjorth Larsen, A. et al. The atomic simulation environment—a Python library for working with atoms. *Journal of Physics: Condensed Matter* **29**, 273002, doi:10.1088/1361-648X/aa680e (2017)). A Langevin thermostat controlled the temperature. First, the molecular geometry was optimized followed by equilibration to 500 K for 2.5 picoseconds with a timestep of 0.25 femtoseconds. Upon equilibration, conformations were sampled every 2 picoseconds from a production run of 200 picoseconds in the NVT ensemble at 500 Kelvin, using a timestep of 0.5 femtoseconds with the same thermostat. This yielded a total of 100 configurations per molecule.

The message passing neural network was based on MACE (Batatia, I. et al. Mace: Higher order equivariant message passing neural networks for fast and accurate force fields. *arXiv preprint arXiv:2206.07697* (2022)). Interatomic distances were incorporated as edge features in the message passing neural network, however, angles were not part of the edge featurization and no universal node was used in this variation of the message passing neural network. A cutoff radius of 5Å was used.

Experimental, SI

11. Compound characterization for P450 and Metallozyme reactions missing in SI (including NMRs). This characterization should be added.

Characterization of these compounds (S25 - S28) has been added.

12. High-resolution mass spectrometry (HRMS) data is missing. HRMS data should be added for isolated, novel, and fully-characterized products.

HRMS data for all compounds in the SI has been added.

Comments on the chosen title:

13. Title is partially misleading. The authors claim to predict Minisci and P450 LSF reactivity. The dataset mostly contains Minisci reactions (and two P450/metalloenzyme reactions of 25 on one drug molecule (Loratadine) in the retrospective test set) - not sufficient to show that the model can accurately predict P450 and metalloenzyme reactions on drug or late-stage molecules. If the inclusion of P450 had mainly the purpose of increasing the performance on Minisci alkylations, it should be treated like this.

Whilst we tested the model on a P450-only test set, we understand your point that the bulk of the validation was done on Minisci conditions. We have changed the title to reflect that.

Predictive Minisci Late Stage Functionalization with Transfer Learning

Reviewer #3 (Remarks to the Author):

The authors present a tool to predict regioselectivity in late-stage C-H functionalization, specifically for radical addition to heterocycles in Minisci reactions.

Before accepting to review, I had indicated that I am not an expert in the tools that are the subject of this manuscript but rather an expert in late stage functionalization method development.

I agree with the authors that there is a current lack of predictability of regioselectivity in the Minisci and related reactions. I would encourage the authors to limit their claims to such reactions, and not generalize to late-stage functionalization in general as several of the claims are only appropriate for the subset of reactions that is discussed here. There are other examples of late-stage functionalization reactions that have other limitations. This is a quick but important fix for the manuscript.

I also concur with the authors that the reaction class that they have selected is an important one. While I cannot comment on the innovation and novelty of the tools developed, I would comment that, even if successful, they are currently only described for one single subclass of a well-defined reaction, which limits the potential scope and utility. This concern may also be rather readily addressable, either by expanding conceptually or by clearly indicating the limitation of scope described here.

With respect to late-stage functionalization chemistry, no conceptual advance is presented here but then I also do not think that that is what is claimed and what the paper is about. Given that the crux of the manuscript is beyond my expertise, I cannot make a recommendation in regards to suitability for the journal but would encourage the authors to consider my comments irrespective of future outlet.

We thank the reviewer for their positive assessment of our work and the contribution in expanding the chemists' toolbox of late stage functionalization methodology development.

In regard to your two concerns, we have revised our manuscript to make it clearer to the reader that we are primarily focused on the *Minisci reactions*, not late-stage functionalizations in general, and have highlighted promising results on P450 enzymatic oxidations, explicitly stating that the scope of the methodology are these two transformations.

As a proof of concept, we highlight our framework's predictive ability on both Baran and Molander-type Minisci and P450 LSFs, transformations whose substrate scope is well defined.

We showcase a transfer learning methodology on ^{13}C NMR shift prediction which boosts zinc sulfinate and BF_3K salt Minisci regiochemical outcome prediction performance above the accurate Fukui-index reactivity scores, and two reactivity prediction machine learning models, on a narrow yet well-defined slice of chemical space. Promising predictive accuracy was also observed on P450 enzymatic oxidations, a chemistry with a broader scope than the aforementioned Minisci conditions, was also achieved. Our ^{13}C NMR data is open-source and we anticipate that the incorporation of larger proprietary ^{13}C NMR datasets as the first step in this transfer learning methodology will expand this methodology to include in other LSF

chemistry.

REVIEWERS' COMMENTS

Reviewer #1 (Remarks to the Author):

This manuscript is a revised version of a submission I reviewed earlier. At that point, my main criticism was that although overall interesting and well done, in the absence of the underlying data the results are not reproducible. The authors have not completely resolved this issue but made a good-faith effort that was approved by the editor, who should be the final judge of these things. The only other comment I would have is that now that the authors have compiled at least part of the dataset, they should consider making it available beyond the SI of the paper, e.g. in a public database such as the Open Reaction Database, which would increase the impact of the paper, and to pursue the publication of the full dataset with their partners.

Regarding the other comments, the authors have again made a good-faith effort to provide additional information (e.g. the Meta-UGT and the including of the AUROC) and to tone down some of their overly broad statements in the previous version of the manuscript. Invariably, this is more successful in some cases than others. For example, Ref 11a specifies that performance is "on par" with trained humans, not "on occasion, surpass them" as stated on p.3. Given that the model in Ref11a predicts the right product in only ~85% of the cases, maybe that says more about the human than the model. Given the already prevalent hype in the field, a nuanced view will serve the field better than unbridled triumphalism and I encourage the authors to do maybe one more editorial pass with this in mind.

Overall, I do think that the manuscript is well done and will find significant interest in the community. I therefore recommend it for publication.

Reviewer #2 (Remarks to the Author):

I thank the authors for their thorough revisions of their manuscript. I especially like the newly added Figure 3 describing the prediction on symmetric molecules.

Missing experimental data (NMRs and HRMS) has been added to the SI. Additionally, descriptions of the network architectures (MPNN_{P450} and MPNN_{LSF}), the MD simulation, the 3DGNN architecture and the 3D molecular representation have been added.

Moreover, newly added Table S1 and Figure S1 give essential additional information about the data set and model performance. Furthermore, the updated title is very suitable.

The authors have addressed all my concerns and questions and I recommend the manuscript for publication in Nature Communications.

Kenneth Atz

Reviewer #1 (Remarks to the Author):

This manuscript is a revised version of a submission I reviewed earlier. At that point, my main criticism was that although overall interesting and well done, in the absence of the underlying data the results are not reproducible. The authors have not completely resolved this issue but made a good-faith effort that was approved by the editor, who should be the final judge of these things. The only other comment I would have is that now that the authors have compiled at least part of the dataset, they should consider making it available beyond the SI of the paper, e.g. in a public database such as the Open Reaction Database, which would increase the impact of the paper, and to pursue the publication of the full dataset with their partners.

We thank the reviewer for their understanding and flexibility and for their suggestion. Our public dataset will be freely available on our GitHub repo (https://github.com/emmaking-smith/SET_LSF_CODE) and in the Open Reaction Database.

Regarding the other comments, the authors have again made a good-faith effort to provide additional information (e.g. the Meta-UGT and the including of the AUROC)) and to tone down some of their overly broad statements in the previous version of the manuscript. Invariably, this is more successful in some cases than others. For example, Ref 11a specifies that performance is “on par” with trained humans, not “on occasion, surpass them” as stated on p.3. Given that the model in Ref1 1a predicts the right product in only ~85% of the cases, maybe that says more about the human than the model. Given the already prevalent hype in the field, a nuanced view with serve the field better than unbridled triumphalism and I encourage the authors to do maybe one more editorial pass with this in mind. Overall, I do think that the manuscript is well done and will find significant interest in the community. I therefore recommend it for publication.

Thank you for assisting us in the editing of this manuscript. We have gone through the manuscript again to tone down subjective language such as "extremely" / "ultimate" / etc. We have revised the sentence in question to:

Some experimentally-based reactivity models can reach human expert performance in their predictions.